# HPLC Analysis and In Vivo Renoprotective Evaluation of Hydroalcoholic Extract of *Cucumis melo* Seeds in Gentamicin-Induced Renal Damage

**DOI:** 10.3390/medicina55040107

**Published:** 2019-04-15

**Authors:** Mohammad Saleem, Fatima Javed, Muhammad Asif, Muhammad Kashif Baig, Mehwish Arif

**Affiliations:** 1Department of Pharmacology, University College of Pharmacy, University of the Punjab, Lahore, Punjab 54000, Pakistan; 2Department of Pharmacology, Faculty of Pharmaceutical Sciences, Government College University, Faisalabad, Punjab 38000, Pakistan; Fatima_javied@yahoo.com (F.J.); mehwisharif61@yahoo.com (M.A.); 3Department of Pathology, Aziz Fatima Medical and Dental College, Faisalabad, Punjab 38000, Pakistan; Baigs23@hotmail.com

**Keywords:** nephroprotective effect, *Cucumis melo* seeds, hydroalcoholic extract, gentamicin, HPLC analysis

## Abstract

*Background and objectives: Cucumis melo*, of family Cucurbitaceae, has traditionally been used to treat variety of kidney disorders. However to best of our knowledge there is no scientific study available that validates its renaoprotective uses. Therefore, this study aimed to evaluate nephroprotective effects of hydroalcoholic extract of *Cucumis melo* seeds (CMHE) and to identify its phytoconstituents. *Materials and Methods*: HPLC was performed to identify key phytochemicals of CMHE. Gentamicin (100 mg/kg/day, i.p) was administered to induce nephrotoxicity in Swiss albino mice for 8 days. Gentamicin (100 mg/kg/day, i.p) and oral CMHE were co-administered to mice at doses of 250 and 500 mg/kg to evaluate protective effects of CMHE. Normal control group mice were administered normal saline. Changes in body weights, biochemical and histopathological studies were conducted to establish nephroprotective effects of CMHE. *Results*: HPLC analysis indicated presence of quercetin, *m*-coumaric acid, gallic acid, chlorogenic acid, and *trans*-4-hydroxy-3-methoxy cinnamic acid in CMHE. Mice treated with CMHE showed significant increase in body weight and decrease in kidney weight as compared with toxic control group. Dose-dependent significant decrease in total blood urea nitrogen, serum creatinine, serum urea, and uric acid levels were observed in CMHE-treated groups as compared with toxic control group. Histopathological analysis of CMHE-treated groups showed improvement in kidney structures as compared with toxic control group. *Conclusions*: Biochemical, histopathological, and phytochemical screening of hydroalcoholic extract of *Cucumis melo* seeds suggest that it has nephroprotective potential. Furthermore, standardization of extract against identified phytochemicals, as well as long-term toxicological studies are suggested before commencement of clinical trials.

## 1. Introduction

Nephrotoxicity, one of the most common renal problems, may occur after exposure to variety of commonly used drugs and/or environmental toxins and can lead to temporary or permeant renal dysfunction [1]. Inflammation and oxidative stress are two major contributors in renal failure [2]. Currently researchers are working to explore protective attributes of new agents using different animal models including aminoglycosides induced-nephrotoxicity model [3]. Among all aminoglycosides, gentamicin is the most widely consumed and studied antibiotic [4]. Incidents of nephrotoxicity are approximately 13–30% in gentamicin-treated patients [5]. Gentamicin stimulates generation of reactive oxygen species, leading to renal damage [6]. It induces generation of hydroxyl radicals and superoxide anions from renal mitochondria [7], resulting in impairment of mitochondrial respiration and cation transport [8].

Herbal use for medicinal purposes is increasing worldwide [9]. The World Health Organization has reported an increasing trend of using herbal medicines in more than 80% of population of developing countries [10]. Researchers have also found various phytochemicals and medicinal plants exhibiting nephroprotective effects like gingerol from *Zingiber officinale*, ginsenosides from *Panax ginseng*, crocin from *Crocus sativus*, and quercetin from *Ginkgo biloba* [2].

Members of family Cucurbitaceae also have great medicinal importance. Many medicinally important plants, such as *Momordica charantia*, *Citrullus lanatus*, *Cucurbita maxima,* and *Citrullus colocynthis* belong to this family [11]. *Cucumis melo* is another member of this family and is commonly known as musk melon and khrbooza in Pakistan. Analgesic, anti-inflammatory, antiulcer, antidiabetic and hepatoprotective activities of *Cucumis melo* have already been reported. It is used traditionally to lower blood pressure and maintain kidney functions [12]. Its seeds have also been used for oliguria, burning micturition, diuretic, and lithotriptic effects [13]. Based on these traditional uses, the present study was carried out to analyze the nephroprotective activity of *Cucumis melo* seeds.

## 2. Material and Method

### 2.1. Collection of Plant Material

*Cucumis melo* specimens were collected in the month of April from Punjab, Pakistan. The sample was identified and authenticated by Dr. Mansoor, taxonomist at Department of Botany, University of Agriculture, Faisalabad, Pakistan. *Cucumis melo* was cut, seeds were separated and thoroughly washed with tap water. Seeds were dried under shade at room temperature until fully dried and were ground into coarse powder using an electric grinder.

### 2.2. Preparation of Extract

Powdered seeds were macerated in 3.5 L ethanol: water (70:30) for 7 days with frequent shaking. Soaked material was filtered, and collected filtrate was evaporated with the help of a rotary evaporator at temperature set at 45 °C to get hydroalcoholic extract of *Cucumis melo* seeds (CMHE).

### 2.3. Preliminary Qualitative Phytochemical Screening

Different tests for the identification of terpenoids, alkaloids, saponins, tannins, flavonoids, and anthraquinone in CMHE were performed following reported methods [14]. Please see the Appendix A for detailed procedures.

### 2.4. Quantitative Phytochemical Analysis by HPLC

Sample was prepared by dissolving 0.2 mg of CMHE in 5 mL of double distilled water. The sample solution was mixed with 15 mL methanol (100%), then shaken well and kept at room temperature for 5 min. Then, 5 mL of double distilled water was added to the mixture with shaking. Again, the solution was allowed to stay undisturbed for 5 min at room temperature. After adding 10 mL of 15 M HCl, the solution was filtered and placed in oven at 95 °C for 2 h. Later, it was filtered using syringe filter and analyzed via HPLC [15]. HPLC with a UV–Visible detector and a Shim-Pack CLC-ODS (C-18) column (25 cm × 4.6 mm, 5 μm) was used for analysis. A (double distilled water: acetic acid ratio of 94:6, pH = 2.27) and B (100% acetonitrile) were used as mobile phase. Mobile phase was run in isocratic mode with a flow rate of 1 mL/min, and each run time was 10 min. Recordings were obtained at 280 nm wavelength using UV-visible detector. Peaks of separated compounds and reference compounds were compared by comparison of their retention time and UV-spectra [16].

### 2.5. Nephroprotective Activity

#### 2.5.1. Experimental Animals

Swiss albino mice were procured from National Institute of Health, Islamabad, weighing 20–25 g and housed in the animal house of Faculty of Pharmaceutical Sciences, Government College University Faisalabad. Mice were kept in polypropylene cages under an ambient temperature of 25 ± 2 °C with a relative humidity of 50 ± 15% and 12 h light and dark cycles [17]. All the animal handling procedures were approved by the Institutional Review Board of Government College University, Faisalabad (Study No. 19599, Reference No. GCUF/ERC/1999).

#### 2.5.2. Study Design

Animals were randomly divided into four groups having six animals in each group (*n* = 6).

Group I: Served as normal control and received normal saline (vehicle for CMHE) for 8 days.

Group II: Served as nephrotoxic control and received gentamicin (100 mg/kg/day, i.p.) for 8 days.

Group III: Received gentamicin (100 mg/kg/day, i.p.) and CMHE (250 mg/kg/day) orally for 8 days.

Group IV: Received gentamicin (100 mg/kg/day, i.p.) and CMHE (500 mg/kg/day) orally for 8 days.

After 24 h of the last dose administration, experimental animals were sacrificed, and blood samples and kidneys from each group were collected [18]. Blood samples were centrifuged at 6000 rpm for 10 min at 25 °C and serum were used for biochemical analysis.

#### 2.5.3. Body and Organ Weight Changes

Changes in body and kidneys weights of mice from all groups were observed.

#### 2.5.4. Biochemical Analysis

Serum urea, blood urea nitrogen, serum creatinine, and uric acid were determined by using Crescent diagnostic (Jeddah, Saudi Arabia) kits.

#### 2.5.5. Histopathology

Kidneys from all groups were harvested and preserved in 10% formalin. Tissues sections of 5 μm thickness were prepared from the paraffin fixed tissue blocks and stained using hematoxylin and eosin stains. Tissue sections were evaluated by an expert pathologist (Dr. Muhammad Kashif Baig) for histopathological changes (i.e., tubular degeneration, interstitial inflammation, and tubular necrosis) in different treatment groups. Damage scoring was as follows: Zero = no damage, Grade 1 = 1–20% damage, Grade 2 = 21–40% damage, Grade 3 = 41–60% damage, Grade 4 = 61–80% and Grade5 = 81–100% damage respectively.

### 2.6. Statistical Analysis

Data were analyzed by two-way ANOVA (analysis of variance) using Bonferroni post-test and results were represented as mean ± SD (standard deviation) (*n* = 6). Comparison was made different parameters of control and CMHE treatment groups and between toxic control group parameters and CMHE treatment groups respectively. A *p* value less than 0.05 was considered statistically significant.

## 3. Results

### 3.1. Preliminary Qualitative Phytochemical Analysis

Data of preliminary phytochemical analysis of CMHE is presented in Table 1.

### 3.2. Quantitative Phytochemical Analysis by HPLC

HPLC analysis of CMHE showed presence of quercetin, gallic acid, chlorogenic acid, *m*-coumaric acid, and *trans*-4-hydroxy-3-methoxy cinnamic acid (Figure 1). Compounds were identified by comparing their retention time with standard/reference compounds (Table 2, Appendix A).

### 3.3. Body and Organ Weight Changes

The final body weights of mice were observed to be significantly decreased (*p* < 0.001) in gentamicin-treated animals (toxic control group) when compared with their respective initial weights. In contrast, non-significant (*p* > 0.05) changes were observed in final weights of animals in normal control and CMHE treatment groups (250 and 500 mg/kg) when compared with their respective initial weights (Table 3). In normal healthy animals, a slight (3%) increase in body weight was observed.

### 3.4. Biochemical Analysis

Serum creatinine levels in toxic control group (gentamicin-induced toxicity) were observed to be significantly increased (*p* < 0.01) to 1.92 ± 0.015 mg/dL when compared with serum creatinine levels in normal control group (i.e., 0.70 ± 0.009 mg/dL). CMHE (250 and 500 mg/kg)-treated groups showed significant decrease in serum creatinine level (0.91 ± 0.017 mg/dL (*p* < 0.05) and 0.82 ± 0.016 mg/dL (*p* < 0.01)) respectively when compared with creatinine levels in toxic control group. Serum urea levels (72.08 ± 1.197 mg/dL) in toxic control group were significantly increased (*p* < 0.001) as compared with serum urea levels (40.33 ± 1.168 mg/dL) in normal control group. *CMHE* administration showed significant decrease (*p* < 0.01) in serum urea levels, with the value of 48.43 ±0.918 mg/dL at 250 mg/kg dose and 45.88 ± 0.920 mg/dL at 500 mg/kg dose, respectively.

Blood urea nitrogen level (BUN) was also observed to be increased significantly in toxic control group (33.68 ± 0.560 mg/dL) as compared with BUN in normal control group (18.84 ± 0.546 mg/dL). Administration of CMHE significantly decreased (*p* < 0.01) level of BUN with values of 22.63 ± 0.429 mg/dL and 21.44 ± 0.430 mg/dL at 250 and 500 mg/kg, respectively, when compared with BUN value (33.68 ± 0.560 mg/dL) in gentamicin-induced toxicity group. A significant increase (*p* < 0.01) in serum levels of uric acid (i.e., 5.12 ± 0.011 mg/dL) was observed in gentamicin-treated toxic group when compared with serum levels of uric acid (i.e., 4.03 ± 0.072 mg/dL) in saline-treated group. A concentration-dependent decrease (*p* < 0.01) in serum levels of uric acid was also observed in mice groups treated with 250 and 500 mg/kg of CMHE (4.05 ± 0.033 and 3.96 ± 0.138 mg/dL, respectively) when compared with uric acid levels (5.12 ± 0.011 mg/dL) in toxic control group (Table 4, Appendix A).

### 3.5. Histopathological Examination

Histopathological analysis of kidneys from different treatment groups are presented in Table 5. Kidney sections of normal control group showed normal histological structures with normal renal parenchyma, tubules, and glomeruli (Figure 2A), while histopathology of kidney sections of toxic control group displayed tubular atrophy, thinning, and edema in renal parenchyma (Figure 2B). In CMHE 250 mg/kg/day treatment group, renal parenchyma showed moderate to severe lymphocytic infiltrate (Figure 2C), while in CMHE 500 mg/kg/day treated group, renal parenchyma showed foci with mild to moderate lymphocytic infiltrate (Figure 2D).

## 4. Discussion

Nephrotoxicity is caused by multiple commonly used drugs. Gentamicin and acetaminophen induce toxicity due to metabolic activation, leading to generation of reactive oxygen species (ROS) and superoxides [19]. Gentamicin toxicity alters morphological and physiological features of kidneys which are characterized by structural modifications of renal tissues and tubules along with elevation in levels of serum biomarkers including urea and creatinine [20]. In the current study, gentamicin (100 mg/kg/day) administration in albino mice displayed toxicity symptoms, as indicated by increase in serum levels of creatinine, urea, BUN, and uric acid, as well as changes in kidney cellular structure (i.e., tubular atrophy, thinning, and edema in renal parenchyma). In a study conducted by Khushboo K. et al., gentamicin 100 mg/kg/day was also shown to induce nephrotoxicity with similar symptoms [21]. Gentamicin-induced toxic effects were observed to be partially reversed in animals treated with hydroalcoholic extract of *Cucumis melo* seeds. CMHE treatment significantly decreased serum levels of creatinine, urea, BUN, and uric acid, indicating nephroprotective potency of *Cucumis melo* seeds. This data is also supported by histopathological analysis indicating marked improvement in tubular texture, especially at the dose of 500 mg/kg dose of CMHE. It is noteworthy that no tubular necrosis was observed in either CMHE treatment group, while severe tubular inflammation and necrosis with moderate tubular degeneration were observed in toxic control group, indicating the induction of nephrotoxicity by gentamicin. Antioxidant activity of *Cucumis melo* in different in vitro models has already been established [22], which is also suggested to be responsible for protection against ROS-induced renal damage in the current study.

Body weight in toxic control group was observed to be decreased, whereas weight of kidneys was increased. The observed decrease in body weight in gentamicin- induced toxicity group have also been reported by other research studies conducted by Tavafi et al. [23] and Harlalka et al. [24]. On the other hand, increase in kidneys weight might be due to tubular necrosis induced by gentamicin leading to edema [25], and is also supported by our histopathological observations. Moreover, non-significant (*p* > 0.05) changes in final weights of animals treated with 250 and 500 mg/kg of CMHE were observed when compared with their respective initial weights. Change in body weight is an important indicator of toxicity, and it has been advocated that if a treatment induces a 10% or greater decrease in body weight in healthy animals, it indicates its toxic nature [26]. In our study, decrease in body weights of animals treated with 250 and 500 mg/kg of CMHE was less than 5%, which indicated a relatively non-toxic nature of plant extract. On the other hand, in toxic control group, 10% decrease in body weight was observed, highlighting the induction of toxicity. This body weight change data also supports the finding of histopathological and biochemical observations of the current study.

HPLC analysis of CMHE revealed presence of potent nephroprotective phytochemicals. Various scientific studies have indicated nephroprotective property of quercetin. It has been proposed to have renal anti-inflammatory effects and has potential to reduce interstitial fibrosis and kidney injury [27]. In another study, quercetin also showed improvement in renal injury due to ischemia–reperfusion [28]. Its antioxidant activity has been suggested to be responsible for protection against lead-induced renal inflammation and oxidative stress [29]. Gallic acid has also been reported to have renal protective activity against doxorubicin-induced chronic kidney disease [30]. Its nephroprotective effects were also observed in nephrotoxicity and oxidative stress induced by sodium fluoride in rats [31]. It also inhibited renal injury and dysfunction in experimentally induced-diabetic nephropathy in rats [32]. Chlorogenic acid has also attenuated kidney injury induced by D-galactose [33] and cisplatin [34] in mice models. *trans*-4-hydroxy-3-methoxy cinnamic acid, also known as ferulic acid, has been shown to protect renal cells from stress and injury induced by glycerol in rats [35]. Its protective role was also observed in diabetic nephropathy model in OLETF rats [36]. However, long-term use of ferulic acid was found to have nephron-damaging property in chronic kidney disorder induced by doxorubicin due to its pro-oxidant activity [30]. In our current study, tubulointerstitial inflammation was observed in both treatment groups, though the extent was lesser compared with toxic control group. Ferulic acid might be responsible for this observation; however, further detailed studies are required to establish this.

Traditionally, seeds of *Cucumis melo* have been used as a diuretic and for the treatment of renal disorders like burning and painful micturition, bladder and kidney stones, renal inflammation, and urinary tract ulcers and infections [13]. A study conducted on ethanolic extract of *Cucumis melo* seed displayed a strong diuretic activity in rats [37]. Therefore, current pharmacological evaluation and its folklore use suggest its development as a nephroprotective agent after further detailed pharmacological and phytochemical investigations.

## 5. Conclusions

The current study highlights the nephroprotective effect of *Cucumis melo* seeds, which is suggested to be due to the presence of multiple antioxidant constituents, leading to improvements in biochemical and histopathological aberrations induced by gentamicin. Moreover, effects on in vivo oxidative and inflammatory markers (e.g., glutathione peroxidase, superoxide dismutase, catalase, and C-reactive protein) should also be evaluated in animal models, as oxidative stress is mainly responsible for renal failure. Furthermore, modern extraction techniques should be adopted to reduce the percentage/amount of ferulic acid in the extract. Overall, *Cucumis melo* seeds extract can be used as an effective herbal remedy for the prevention and treatment of a variety of kidney disorders.

## Figures and Tables

**Figure 1 medicina-55-00107-f001:**
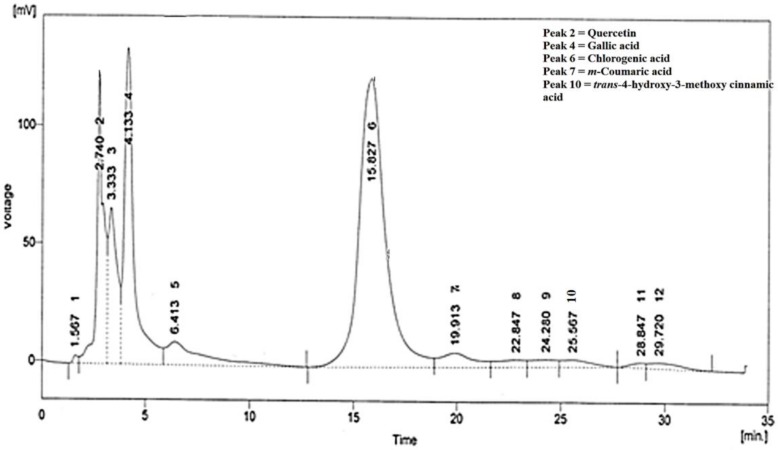
HPLC chromatogram of CMHE. Compounds were identified by comparing retention times with standard compounds.

**Figure 2 medicina-55-00107-f002:**
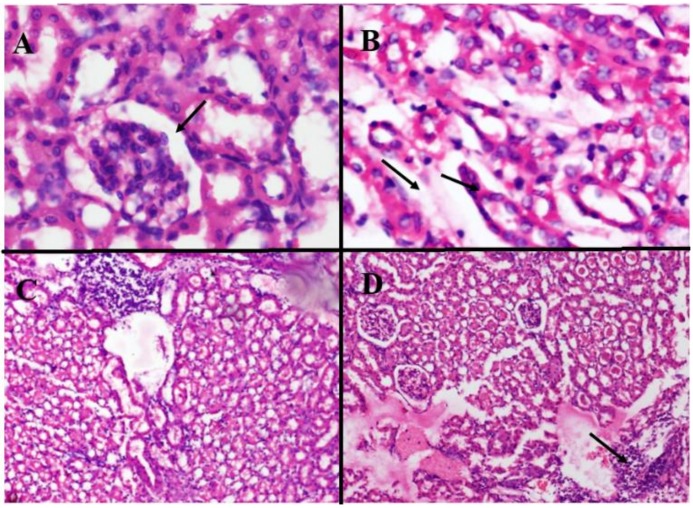
Histological evaluations of kidney sections of the (**A**) normal control group showing normal renal parenchyma with normal tubules and glomeruli; (**B**) Toxic control group representing renal parenchyma with tubular atrophy, thinning, and edema; (**C**) CMHE 250 mg/kg/day treatment group showing renal parenchyma with moderate to severe lymphocytic infiltrate; (**D**) CMHE 500 mg/kg/day treatment group showing renal parenchyma with foci of mild to moderate lymphocytic infiltrate. Photos were taken at 40× magnification.

**Table 1 medicina-55-00107-t001:** Qualitative phytochemical screening of hydroalcoholic extract of *Cucumis melo* seeds (CMHE).

	Tannins	Alkaloids	Flavonoids	Saponins	Anthraquinone	Terpenoids
*Cucumis melo*	+	+	+	++	−	++

Where: + = slight presence, ++ = maximum presence, − = absence.

**Table 2 medicina-55-00107-t002:** HPLC analysis of hydroalcoholic extract of *Cucumis melo* seeds (CMHE).

Compound	Retention Time (min)	Area	Area (%)	Quantity (ppm)
Quercetin	2.740	2742.368	11.8	145.3
Gallic acid	4.133	4444.478	19.1	159.9
Chlorogenic acid	15.827	10,852.407	46.7	846.4
*m*-Coumaric acid	19.913	691.188	3.0	8.29
*trans*-4-Hydroxy-3-methoxy cinnamic acid	25.567	298.775	1.3	10.43

**Table 3 medicina-55-00107-t003:** Effect of CMHE on body weight and kidney weight.

Treatment Groups	Initial Body Weight (g)	Final Body Weight (g)	Body Weight Change (g)/%	Kidney Weight (g)
Normal Control	22.00 ± 0.40	22.73 ± 0.32 ^ns^	0.73 ± 0.13 (3.31%)	0.6 ± 0.09
Toxic Control (Gentamicin 100 mg/kg/day)	22.00 ± 0.40	19.78 ± 0.46 ***	−2.23 ± 0.22 (−10.03%)	0.73 ± 0.09
Gentamicin + CMHE 250 mg/kg/day	23.75 ± 0.75	23.03 ± 0.69 ^ns^	−0.73 ± 0.08 (−3.0%)	0.63 ± 0.05
Gentamicin + CMHE 500 mg/kg/day	23.68 ± 0.34	22.95 ± 0.36 ^ns^	−0.73 ± 0.08 (−3.01%)	0.61 ± 0.04

Data expressed as mean ± S.D. *n* = 6, where, ns = non-significant, *** *p* < 0.001, and − sign indicates a decrease in body weight.

**Table 4 medicina-55-00107-t004:** Biochemical parameters studied to evaluate nephroprotective effects of CMHE.

Treatment Groups	Creatinine (mg/dL)	Urea (mg/dL)	BUN (mg/dL)	Uric Acid (mg/dL)
Normal Control	0.70 ± 0.009	40.33 ± 1.168	18.84 ± 0.546	4.03 ± 0.072
Toxic Control (Gentamicin 100 mg/kg/day)	1.92 ± 0.015 **	72.08 ± 1.197 ***	33.68 ± 0.560	5.12 ± 0.011 **
Gentamicin + CMHE 250 mg/kg/day	0.91 ± 0.017 ^#^	48.43 ± 0.918 ^###^	22.63± 0.429 ^###^	4.05 ± 0.033 ^##^
Gentamicin + CMHE 500 mg/kg/day	0.82 ±0.016 ^b^	45.88 ± 0.920 ^c^	21.44 ± 0.430 ^c^	3.96 ± 0.138 ^b^

Values expressed as mean ± S.D. *n* = 6. Asterisks indicate normal control vs. toxic control; ^#^ sign indicate toxic control vs. gentamicin + CMHE 250 mg/kg; ^a,b,c^ letters indicate toxic control vs. gentamicin + CMHE 500 mg/kg. ^a,#^
*p* < 0.05, ^b,##,^** *p* < 0.01, ^c,###,^*** *p* < 0.001.

**Table 5 medicina-55-00107-t005:** Histopathological evaluation for nephroprotective effects of CMHE.

Treatment Groups	Tubular Degeneration	Tubulo-Interstitial Inflammation	Tubular Necrosis
Normal Control	0	0	0
Toxic Control (Gentamicin 100 mg/kg/day)	2	4	4
Gentamicin + CMHE 250 mg/kg/day	1	4	0
Gentamicin + CMHE 500 mg/kg/day	0	2	0

Values presented are on the basis of histopathological observations. Grade 0 = no damage, grade 1 = mild cellular degradations, grade 2 = moderate cellular degenerations, and grade 4 = severe deformities.

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
