# Peer review of "HPLC Analysis and In Vivo Renoprotective Evaluation of Hydroalcoholic Extract of Cucumis melo Seeds in Gentamicin-Induced Renal Damage"

_medicina, 2019, doi:10.3390/medicina55040107_

Round 1
Reviewer 1 Report
The manuscript is of interest for readers but surely not entirely original. I have some suggestions to increase its impact.
Why females?
It is no clear as saline (group I) was administered
What is the rationale to use 100 mg/kg/day of gentamicin?
How was CMHE administered?
Table 2: the statistical significance to what it refers? To the control group, to the group treated with gentamicin? Are there no statistical differences between the "toxic group" and the control group?
Table 3: I have the same suggestions expressed for Table 2.
The descriptive part of the results must surely be improved, correctly indicating the statistical significance
Author Response
Q1: Why females?
Ans. Corrected. Please refer to section 2.4.1, line 82.
Q2: It is no clear as saline (group I) was administered
Ans. Normal saline was used to dissolve plant extract. That’s why it was administered as in Normal control group.
Q3: What is the rationale to use 100 mg/kg/day of gentamicin?
Ans. Gentamicin at this dose has been shown to induce nephrotoxicity in animals as reported by other research reports. Please refer to line 184-185.
Q4: How was CMHE administered?
Ans. It was administered orally. Please refer to line 91-93.
Q5: Table 2: the statistical significance to what it refers? To the control group, to the group treated with gentamicin? Are there no statistical differences between the "toxic group" and the control group?
Ans. Table is correct please refer to table 3 for details.
Q6: Table 3: I have the same suggestions expressed for Table 2.
Ans. Table is correct please refer to table 4 for details.
Reviewer 2 Report
The aim of this study is to evaluate nephroprotective effects of hydroalcoholic extract of Cucumis melo seeds (CMHE) and to identify its nephroprotective constituents. The author showed that the serum creatinine and BUN are decreased coadministration of CMHE.
However, there are several major points should be revised for publication.
Line 105: What is standard kit?
Line 110: The author should describe which post hoc test uses.
Tabel 1: How to class these parameters?
Line 118-119 and Fig 1: How to determine these peaks is depend on these compounds?
Line 128: Table 2 is 3. Same mistake is available.
Table 3: The BUN of Gentamicin+CMHE 500mg/kg is very low. Why is this parameter low compared to control?
Table 4: How to class these histopathological changes.
Fig 2: The author don’t describe how to stain the kidney section. Moreover, the magnification and section are not same in these panel. This evaluation is unsuitableness.
Author Response
However, there are several major points should be revised for publication.
Q1: Line 105: What is standard kit?
Answer: Serum urea, blood urea nitrogen, serum creatinine and uric acid were determined by using Crescent diagnostic (Jeddah) kits. Kits used have been mentioned. Please refer to line 101-102.
Q2: Line 110: The author should describe which post hoc test uses.
Answer: Data was analyzed by two-way ANOVA (analysis of variance) using Bonferroni post-test and results were represented as mean ± SD (standard deviation) (n = 6).. Please refer to line 109-110.
Q3: Tabel 1: How to class these parameters?
Answer: Qualitative assays were used to detect the presence of these classes of phytoconstituents.
Q4: Line 118-119 and Fig 1: How to determine these peaks is depend on these compounds?
Answer: These compounds were identified by comparing the retention time with that of standard compounds. Please refer to supplementary figure S1.
Q5: Line 128: Table 2 is 3. Same mistake is available.
Answer: Corrected as suugested. Please refer to line 130.
Q6: Table 3: The BUN of Gentamicin+CMHE 500mg/kg is very low. Why is this parameter low compared to control?
Answer: Typo error, corrected accordingly. Please refer to Table 4.
Q7: Table 4: How to class these histopathological changes.
Answer: These changes were classed based on histopathological observations.
Q8: Fig 2: The author don’t describe how to stain the kidney section. Moreover, the magnification and section are not same in these panel. This evaluation is unsuitableness.
Answer. Kidneys were collected and preserved in 10% formalin. Tissues section of 5 μm thickness were prepared from the paraffin blocks and stained using haematoxylin and eosin stains. Tissue sections were evaluated by expert pathologist (Dr. Muhammad Kashif Baig) for histopathological changes i.e., tubular degeneration, interstitial inflammation and tubular necrosis in different treatment groups. Please refer to line 104-109.
Round 2
Reviewer 1 Report
The authors answered fully to my criticisms
Author Response
Appreciate you efforts to review this manuscript.
Kind regards!
Reviewer 2 Report
Line 66-67: The author described that “Different tests for the identification of terpenoids, alkaloids, saponins, tannins, flavonoids and anthraquinone in CMHE were performed following reported methods [14].” However, there are no detailed explain about Phytochemical Screening in this reference. The author must write how to do this experiment.
Line 106-111: The author must explain how to class these histopathological changes. We can not understand that which level is severe and which level is moderate.
Line 161: Figure2 is unnecessary
Figure3. Photos were taken at 40x. However, the size of nuclei is difference between A,B and C,D. Why is the size of nuclei different?
Author Response
Q1: Line 66-67: The author described that “Different tests for the identification of terpenoids, alkaloids, saponins, tannins, flavonoids and anthraquinone in CMHE were performed following reported methods [14].” However, there are no detailed explain about Phytochemical Screening in this reference. The author must write how to do this experiment.
Ans: Detailed procedures are added as suggested. Please refer to supplementary section for details
Q2: Line 106-111: The author must explain how to class these histopathological changes. We can not understand that which level is severe and which level is moderate.
Ans: Grading on the basis of severity is added in section 2.4.5, lin 109-110 and in table 5.
Q3: Line 161: Figure2 is unnecessary
Ans: Shifted to supplementary file
Q4: Figure3. Photos were taken at 40x. However, the size of nuclei is difference between A,B and C,D. Why is the size of nuclei different?
Ans: Changes in size of cells might be due to deleterious effects of gentamicin and due to infiltration by lymphocytes